# Resistin-like Molecule α and Pulmonary Vascular Remodeling: A Multi-Strain Murine Model of Antigen and Urban Ambient Particulate Matter Co-Exposure

**DOI:** 10.3390/ijms241511918

**Published:** 2023-07-25

**Authors:** Nedim Durmus, Wen-Chi Chen, Sung-Hyun Park, Leigh M. Marsh, Sophia Kwon, Anna Nolan, Gabriele Grunig

**Affiliations:** 1Division of Environmental Medicine, Department of Medicine, New York University Grossman School of Medicine (NYUGSoM), New York, NY 10016, USA; nedim.durmus@nyulangone.org (N.D.); chenw10660116@gmail.com (W.-C.C.); sunghyun.park@nyulangone.org (S.-H.P.); anna.nolan@nyulangone.org (A.N.); 2Division of Pulmonary, Critical Care and Sleep, Department of Medicine, New York University Grossman School of Medicine (NYUGSoM), New York, NY 10016, USA; sophia.kwon@nyulangone.org; 3Ludwig Boltzmann Institute for Lung Vascular Research, Otto Loewi Research Centre, Division of Physiology and Pathophysiology, Medical University of Graz, 8010 Graz, Austria; leigh.marsh@medunigraz.at

**Keywords:** resistin-like molecule, pulmonary hypertension, type 2 inflammation, adaptive immune response, retnla, retnlb, retnlg, mouse strains, experimental pulmonary hypertension, urban PM, urban fine dust, immune response in pulmonary hypertension

## Abstract

Pulmonary hypertension (PH) has a high mortality and few treatment options. Adaptive immune mediators of PH in mice challenged with antigen/particulate matter (antigen/PM) has been the focus of our prior work. We identified key roles of type-2- and type-17 responses in C57BL/6 mice. Here, we focused on type-2-response-related cytokines, specifically resistin-like molecule (RELM)α, a critical mediator of hypoxia-induced PH. Because of strain differences in the immune responses to type 2 stimuli, we compared C57BL/6J and BALB/c mice. A model of intraperitoneal antigen sensitization with subsequent, intranasal challenges with antigen/PM (ovalbumin and urban ambient PM_2.5_) or saline was used in C57BL/6 and BALB/c wild-type or RELMα−/− mice. Vascular remodeling was assessed with histology; right ventricular (RV) pressure, RV weights and cytokines were quantified. Upon challenge with antigen/PM, both C57BL/6 and BALB/c mice developed pulmonary vascular remodeling; these changes were much more prominent in the C57BL/6 strain. Compared to wild-type mice, RELMα−/− had significantly reduced pulmonary vascular remodeling in BALB/c, but not in C57BL/6 mice. RV weights, RV *IL-33* and RV *IL-33*-receptor were significantly increased in BALB/c wild-type mice, but not in BALB/c-RELMα−/− or in C57BL/6-wild-type or C57BL/6-RELMα−/− mice in response to antigen/PM_2.5_. RV systolic pressures (RVSP) were higher in BALB/c compared to C57BL/6J mice, and RELMα−/− mice were not different from their respective wild-type controls. The RELMα−/− animals demonstrated significantly decreased expression of RELMβ and RELMγ, which makes these mice comparable to a situation where human RELMβ levels would be significantly modified, as only humans have this single RELM molecule. In BALB/c mice, RELMα was a key contributor to pulmonary vascular remodeling, increase in RV weight and RV cytokine responses induced by exposure to antigen/PM_2.5_, highlighting the significance of the genetic background for the biological role of RELMα.

## 1. Introduction

Pulmonary hypertension (PH) not only has a high mortality but also has limited treatment options [1,2,3,4,5,6,7,8,9]. Our prior work has shown that pulmonary arterial remodeling and subsequent increased right ventricular (RV) systolic pressure (RVSP) could be induced by Th2 response to soluble antigen and could be exacerbated by urban particulate matter (PM_2.5_) [1,10]. Specifically, CD4+ T cells, B cells, antigen-specific antibody and Interleukins (IL)-13 and IL-17A had critical roles in mediating the severe pulmonary arterial remodeling and PH in our C57BL/6 murine model [1,11,12].

Resistin-like molecule α (RELMα) is an intriguing cytokine at the intersection of the response to hypoxia [13,14,15,16,17] and adaptive immune response and belongs to an ancient mediator family, with resistin being the oldest member [18,19]. In the context of our prior findings, our aim was to identify the mediator down-stream of the adaptive immune response that would induce the PH phenotype. We focused on RELMα because of the intriguing role of this cytokine at the intersection of the immune and the hypoxia response. RELMα was termed “found in inflammatory zone 1” (FIZZ1) because it was originally discovered in inflammatory zones associated with an experimental allergic airway disease model [14,20]. Subsequently, RELMα was identified as a biomarker in Th2 asthma models [21], since it is highly upregulated in epithelial cells and alternatively activated macrophages of the M2 type [22,23]. RELMα was also termed hypoxia-induced mitogenic factor (HIMF) because it is highly upregulated in lung tissues exposed to hypoxia, specifically in vascular smooth muscle cells and endothelial cells [14,16,24,25]. HIMF induces cell proliferation and chemotaxis in smooth muscle cells [13,17,26,27,28]. RELMα belongs to the resistin family of cytokines, which in mice consists of four members: resistin, RELMα, -β and -γ [20,29,30,31].

Mechanistically, RELMα has a critical role in hypoxia-induced PH phenotype [16]. With respect to inflammation induced by an adaptive immune response, RELMα has been reported to have either pro-inflammatory [22,32], anti-inflammatory [23,33] or neutral [34] roles depending on the experimental model used.

The investigation of mediators, like RELMα, that have mechanistic roles in the pathways of hypoxia or immune-response-driven inflammation for the PH phenotype is important because drugs targeting these types of mediators are expected to have an important future role in PH therapy. The current major therapeutic approaches are through inhibitors of the endothelin system, prostanoids or stimulation of soluble guanylate cyclase affecting nitric oxide signaling [35]. These drugs have vasodilative and some anti-proliferative effects [35], and endothelin-1 in the endothelium controls the influx of inflammatory cells [36] as well as transition of acute to chronic kidney injury including inflammation [37]. However, for many types of PH, these drugs do not target the pathogenic processes that cause pulmonary vascular injury and PH [35]. Furthermore, they are not effective in the majority of persons affected by heart-disease-associated PH, WSPH2, or lung-disease-associated PH, WSPH3. This is why we undertook a careful study of the role of RELMα in PH induced by a common urban airborne exposure, antigen–urban particulate matter (PM_2.5_ collected from New York City air).

We used mice deficient in RELMα^−/−^ and measured right ventricular systolic pressure, right heart weight as a measure of hypertrophy, pulmonary arterial remodeling and several mediators of immune and vascular responses in the lungs and right ventricle. We compared C57BL/6 and BALB/c mice because BALB/c mice have relatively larger constrictive responses in the airways compared with C57BL/6 mice [38,39]. Further, the lung’s immune response in C57BL/6 mice is more the pleiotropic T-helper type, while BALB/c mice have a more polarized, type 2 dominant response to antigen exposure [40,41]. This design allowed us to achieve the aim of our study, to understand the function of RELMα for the PH phenotype induced by an adaptive immune response in the lungs.

## 2. Results

Expression of *RELMβ* and *RELMγ* in RELMα^−/−^ mice. All three members of the resistin-like molecule family are located on mouse chromosome 16 in the gene order *retnlb, retnla* and *retnlg*. To test the idea that the gene-deletion manipulation in the RELMα^−/−^ mice could have changed the expression of *RELMβ* and *RELMγ*, we determined the expression of all three resistin-like molecules in the lung and right heart tissues of wild-type and KO mice. Figure 1A shows that antigen and PM_2.5_ exposure significantly increased the expression of *RELMα, β* and γ in the lungs of WT mice, and that this increase occurred to a larger fold-degree in BALB/C strain mice. Figure 1A also shows that RELMα^−/−^ mice of both strains had no *RELMα* mRNA expression and significantly decreased expression of *RELMβ* and *RELMγ* in the lungs compared to wild-type mice.

Expression of *RELMα* and *RELMγ* was readily detected in the right ventricle (RV), as shown in Figure 1B. *RELMβ* was not detectible in the right ventricle. Exposure with antigen/PM_2.5_ significantly increased the expression of *RELMα* in the RV of BALB/c WT mice and *RELMγ* in the RV of WT mice from both strains. As in the lungs, the right ventricles of RELMα^−/−^ mice of both strains had no RELMα expression and significantly decreased expression of *RELMγ* when compared to WT mice, as shown in Figure 1.

In BALB/c KO mice, decreased RELM expression attenuated pulmonary vascular remodeling induced by exposure to antigen and PM_2.5_. Groups of WT and KO mice were given antigen and ambient PM_2.5_ (OVA&PM). As expected, [1,10,11,12,42] WT C57BL/6 and BALB/c mice developed pulmonary vascular remodeling compared to saline, as shown in Figure 2A,C. Figure 3 shows the vascular pathology corresponding to the histology scores. Figure 2E shows representative photomicrographs of lungs from each of the groups of mice. RELMα KO mice of the BALB/c strain had significantly ameliorated vascular remodeling compared to wild type, as shown in Figure 2C. In contrast, RELMα^−/−^ mice of the C56BL/6 strain developed pulmonary vascular remodeling to the same extent as WT, as shown in Figure 2A.

BALB/c RELMα KO mice did not develop severe pulmonary arterial remodeling upon exposure to antigenPM_2.5_. Following exposure to antigen/PM_2.5_, the frequency of severely remodeled arteries was higher in C57BL/6 mice when compared to BALB/c mice, Table 1. In keeping with the data shown in Figure 2C, the percentage of severely remodeled pulmonary arteries was significantly lower in antigen/PM_2.5_-exposed RELMα^−/−^ BALB/c mice when compared to WT, as shown in Table 1. In contrast, WT and RELMα^−/−^ C57BL/6 mice had a similar frequency of severely remodeled arteries in the lungs, as shown in Table 1.

RELMα is not necessary for increases in RV systolic pressure induced by exposure to antigen and PM_2.5_. To understand the role of RELMα for the increase in RV systolic pressures (RVSP), groups of WT and KO mice were given antigen and ambient PM_2.5_ (OVA&PM). As expected, [1,10,11,12,42] OVA&PM exposed WT C57BL/6 and BALB/c mice developed significantly increased RVSP compared to control, as shown in Figure 2B,D. Unexpectedly, RELMα^−/−^ mice of both strains showed increases in RV systolic pressure upon challenge with OVA&PM similar to the WT controls, as shown in Figure 2B,D.

Right heart weights and expression of *ANP*, *IL-33* and *IL-33*-receptor (*ST2*) mRNA in the ventricles of the right heart of WT and RELMα^−/−^ mice. In saline-treated mice, right heart weights were similar in wild-type and RELMα KO mice in both C57BL/6 and BALB/c strains, as shown in Figure 4A,B. However, the right heart weight was higher in BALB/c strain control mice when compared to C57BL/6 control mice. Upon exposure to OVA&PM, only BALB/c WT mice developed significantly increased right heart weights, as shown in Figure 4B. Antigen- and PM_2.5_-exposed C57BL/6 strain wild-type mice showed a trend of increased right heart weights, as shown in Figure 4A. Importantly, RELMα^−/−^ BALB/c mice did not develop significant increases in right heart weight following exposure to OVA&PM, as shown in Figure 4B, while RELMα^−/−^ C57BL/6 mice were not different from wild type, as shown in Figure 4A.

The mRNA expression of *ANP (natriuretic peptide type A)* was measured as an indicator of RV stress [43], as shown in Figure 4C,D. Only BALB/c WT mice, but not RELMα^−/−^ BALB/c mice, developed increased *ANP* expression in the RV following OVA&PM exposure compared to control groups, as shown in Figure 4D. In contrast, the expression of *BNP (natriuretic peptide type B)* mRNA showed trends to increase in C56BL/6 strain mice following exposure to OVA&PM, but not in BALB/c strain mice, as shown in Figure 4E. The mRNA expression of *IL-33* and the *IL-33-receptor ST2* was measured in right heart tissue because *IL-33* and ST2 are thought to have important roles in regulating right ventricular homeostasis [44]. Antigen and PM_2.5_ exposure did not change right ventricular *IL-33* or *ST2* expression in C57BL/6 mice, as shown in Figure 4G,I. In contrast, BALB/c WT but not BALB/c RELMα^−/−^ mice had significantly increased right ventricular *IL-33* expression following antigen/PM_2.5_ exposure, as shown in Figure 4H. The *IL-33* receptor, *ST2*, was expressed at lower levels in the right ventricles of RELMα^−/−^ BALB/c mice, which reached statistical significance in OVA&PM-challenged RELMα^−/−^ BALB/c mice relative to WT, as shown in Figure 4J.

## 3. Discussion

Our data showed that endogenous RELMα was not necessary for the development of increased right ventricular systolic pressures in mice exposed to antigen and PM_2.5_. This conclusion is based on our studies using mice of two different strain backgrounds, C57BL/6 and BALB/c. Based on previous reports showing that RELMα, also called HIMF, was both necessary and sufficient to cause the hypoxia-induced increase in right ventricular systolic pressure [14,15,16,24], our data were unexpected. However, the discrepancy can be explained by the very different experimental systems used.

In our studies comparing WT and RELMα^−/−^ mice, we found distinct differences between the C57BL/6 and BALB/c strains. In the C57BL/6 strain, RELMα was redundant. In the BALB/c strain relative to WT, RELMα^−/−^ mice challenged with antigen and PM_2.5_ did not develop increased right ventricle weight, had ameliorated pulmonary vascular remodeling, and no increase in right ventricle *ANP* expression, together with decreased *IL-33* and *ST2* expression in the right ventricle.

Cardiac myocytes, fibroblasts and endothelial cells express *IL-33* constitutively [44]. *IL-33* expression is further increased by inflammatory stimuli in cardiac myocytes and fibroblasts. Cardiac endothelial cells express the *IL-33*-receptor, ST2, while cardiac myocytes and fibroblasts have a relatively low level of surface ST2 expression [44]. The significantly lower expression of the mRNA of these mediators of inter-cellular communication in the right ventricle of RELMα^−/−^ BALB/c mice may have prevented an increase in right ventricular weight in response to antigen and PM_2.5_ exposure despite the right ventricular systolic pressures that were increased to wild-type levels.

The RELMα^−/−^ mice that we studied also had significantly decreased *RELMβ* and *RELMγ* mRNA expression in the lungs and right ventricle. Therefore, our study indicates a critical role of these mediators in BALB/c mice for the responses to antigen and PM_2.5_, while they were redundant in C57BL/6 mice. The strain difference could be due to differences in the responsiveness of the pulmonary vasculature to proliferative and constrictive cues. In this respect, it is remarkable that C57BL/6 mice have polymorphisms (SNPs) in the 3-prime region of the *BMPR2* (*bone morphogenetic protein receptor, type II*) gene that are not found in BALB/c mice (https://www.informatics.jax.org/snp/marker/MGI:1095407; accessed on 15 May 2023). Decreased function of the BMPR2 gene is associated with a high risk of developing pulmonary hypertension [45]. Another possibility is that potential receptors for the RELM molecules, such as Toll-like receptor (TLR4) or RAGE (advanced glycosylation end-product) [46,47], are polymorphic and differ between C57BL/6 and BALB/c mice. There is strong evidence that resistin binds TLR4 [48,49]; however, little is known about the binding of RELM molecules to TLR4. Further, TLR4 is polymorphic between BALB/c and C57BL/6 mice [50], and this is particularly important because the antigen (OVA) that we used contains lipopolysaccharide, which induces inflammation via TLR4. The RAGE gene is located in the MHC locus, H2K, and this locus is highly polymorphic between BALB/c (H2Kd) and C57BL/6 (H2Kb) mice.

Future investigations need to be designed to understand the interactions of RELM cytokines with the mediators currently targeted by drugs approved to treat pulmonary arterial hypertension and by therapies in development. Of particular interest are agonists of BMPR2 receptor signaling [51] because of their inflammation inhibitory activity [52]. Epigenetic markers like micro-RNA or long-non-coding (lnc) RNA are expected to become biomarkers to identify PH endotypes [53]. Combining deep clinical phenotyping and multi-OMICS analysis of whole blood, and single-cell analysis including genetics and gene-expression, metabolomics [54,55] is expected to generate the knowledge of how to combine currently approved therapies and add potential novel therapeutic targets such as RELMα. Based on the knowledge gleaned from the use of biologics targeting type 2 cytokines in asthma, carefully designed and analyzed, long-duration clinical studies will be required to understand how the use of these biologics might result in the reduction of background standard therapy (for asthma, corticosteroids and bronchodilators) [56].

There are some limitations of our study: (1) First of all, humans only have a single resistin-like molecule, RELMβ, and human RELMβ is induced in hypoxia and is mitogenic for vascular smooth muscle cells [57]. It is thought that mouse RELMα is the homologue of human RELMβ, but mouse RELMα may not have the same function as human RELMβ. (2) Secondly, the RELMα^−/−^ mice of both strain backgrounds that were studied by us have a compound KO phenotype with a deletion of *RELMα*, and significantly depressed *RELMβ* and *RELMγ* mRNA. In mice, the three resistin-like molecules developed by gene-duplication on chromosome 16 and the compound deficiency is likely due to the deletion of a master regulator of all three resistin-like molecules in the KO animals. The compound KO phenotype of our mice may partially explain the different data with respect to the role of RELMα for the pulmonary hypertension phenotype that has been reported in the hypoxia system [15]. Those studies were conducted with *RELMα*-specific inhibitors in wild-type animals. (3) Lastly, our studies used a very low dose of PM_2.5_ that does not induce a response intranasally, which is different from mice given saline intranasally [10]. Therefore, the RELMα-dependent responses identified in our studies (e.g., BALB/c strain mice pulmonary arterial remodeling, increased right ventricular weights and mediator expression) were driven by the adaptive immune response to OVA and perhaps the associated lipopolysaccharide. Future dose–response studies are necessary to understand the role of RELMα in cardiovascular and lung responses to PM_2.5_ exposure.

## 4. Materials and Methods

Ethics Statement. All animal experiments were performed according to guidelines outlined by the United States Department of Agriculture and the American Association of Laboratory Animal Care under the supervision and specific approval of the Institutional Animal Care and Use Committees at New York University, Grossman School of Medicine (IACUC #140812-01).

Mice. RELMα^−/−^ on a C57BL/6 background (Regeneron Pharmaceuticals, Inc., Tarrytown, NY, USA) were backcrossed to BALB/c (Dr. Marc E. Rothenberg’s laboratory, Cincinnati Children’s Hospital Medical Center) and a breeding pair was kindly provided [22,32]. RELMα^−/−^ mice (KO) were then backcrossed to C57BL/6J for 6 generations. C57BL/6J and BALB/c wild-type (WT) mice were purchased (Jackson Laboratory) and adjusted to the environment in our colony for 10 days. Either these mice or littermates of the backcross strain were used for comparison with RELMα^−/−^. Male and female mice 5–7 weeks of age at the start of the experiment were randomized into cages holding up to 4 mice. All mice were housed under pathogen-free conditions.

RELMα^−/−^ Genotyping. A PCR-based method using ear tissues was used for genotyping. Briefly, for pretreatment of ear tissues, 180 µL of 50 mM NaOH was added into a tube containing ear tissues and incubated at 95 °C for 13 min, and stayed at 22 °C until use. 20 µL of 1 M Tris-HCl (pH 8.0) was added, mixed with vortexing. Next, 5 µL of pretreated sample was used for total 25 µL of PCR reaction. TERRA™ PCR Direct Polymerase mix kit (Clontech, Mountain View, CA, USA) was used with primers 5-GTCAGCAATCCCATGGCGTA-3 (forward) and 5-GTCTGTCCTAGCTTCCTCACTG-3 (reverse); 400 bp for KO allele or primers 5-GTCAGCAATCCCATGGCGTA-3 (forward) and 5-ACTTCCCTACCC ACCCATTCC-3 (reverse); 800 bp for WT allele. A gradient PCR method was used with the following conditions: 98 °C for 2 min, followed by 2 cycles of 94 °C for 10 s, 64 °C for 30 s and 72 °C for 50 s, followed every 2 cycles with 1 °C reduced annealing temperature until annealing temperature was 58 °C, followed by 27 cycles of 94 °C for 35 s, 58 °C for 30 s and 72 °C for 50 s, followed by 1 cycle of 72 °C for 10 min, and followed by a hold at 4 °C. The product was run on a gel to distinguish WT, heterozygous and KO mice.

Urban PM (PM_2.5_) (<2.5 mm in aerodynamic diameter) was collected from New York ambient air and resuspended as previously described [58,59]. PM_2.5_ was diluted in phosphate-buffered saline (PBS), ultrasonicated before use and mixed with OVA solution so that the final concentration was 25 μg PM_2.5_/50 µL intranasal dose.

Antigen priming, antigen and PM_2.5_ challenge (OVA and PM). Animals were primed and challenged with antigen as previously published [1,10,11,12,42]. Briefly, mice were injected intraperitoneally with Ovalbumin (OVA) (grade V; Sigma-Aldrich, St. Louis, MO, USA; 50 µg/dose) adsorbed to Alum (Imject Alum; Thermo Fisher Scientific, Rockford, IL, USA; 2 mg/dose) at a two-week interval. Two weeks later, the mice were intranasally challenged with either PBS or combined OVA (100 µg/dose) and PM_2.5_ (25 μg/dose) 2 times each week, for a total of 6 doses given over a three-week period. The mice were analyzed one day following the last intranasal exposure, as shown in Figure 5. This OVA preparation contained lipopolysaccharide (LPS), contributing to the responses in the airways and lungs to intranasal OVA challenge in this mouse model [60].

Group comparisons. In a pilot study, we compared sensitized C57BL/6 or BALB/c wild-type or KO mice intranasally exposed to either saline, PM2.5, OVA or OVA&PM2.5. The data of the pilot study showed that, as expected [10], all groups of saline- or PM_2.5_-exposed mice showed no significant difference in any of our measurements. Therefore, we pooled these two groups. Further, the pilot study showed that the comparison of the responses of WT and KO mice to OVA alone were the same as when WT and KO mice were exposed to OVA and PM_2.5_. For these reasons, the data from these two exposure groups were pooled to increase statistical power for the WT vs. KO comparisons.

RV Systolic Pressure (RVSP). RVSP was measured by inserting a catheter via the jugular vein in anaesthetized, spontaneously breathing mice [10,42,61]. Mice were analyzed without prior knowledge of group identity, and we alternated mice from different cages to eliminate cage effects. Briefly, mice were anaesthetized by intraperitoneal injection of Avertin (Sigma-Aldrich) made by mixing 5 mL of 2-Methyl-2-butanol and 5 g of 2,2,2-Tribromoethanol; 0.25 mL of the stock solution was diluted with 10 mL of saline solution; the mice were injected according to their weight, 10 μL/g mouse. Once surgical plane of anesthesia was reached, the right jugular vein was isolated. The pressure catheter (F1.4, Millar Instruments, Inc. Houston, TX, USA) was inserted and advanced into the RV to measure the pressure. Then, the mice were euthanized with an overdose of barbiturate and lung and heart tissues were harvested. The RV pressure data were analyzed using the LabChart 7 program (ADInstruments, Colorado Springs, CO, USA).

RV Hypertrophy, Fulton’s Index [right/(left + septum) ventricular weight]. The RV and left ventricle plus septum were removed and weighed. The data were used to calculate the right ventricular weight relative to the weight of the left ventricle and septum.

Pulmonary vascular remodeling quantification. The approach was as previously described [1,62,63,64,65] and the analysis was performed without knowledge of the group identity of the sample by scoring. At least 20 consecutive view fields were randomly selected in all available lung lobes and were evaluated at 200× magnification. All arteries with a diameter of <100μm were scored (1—normal to 4—severe remodeling, Figure 3). For each lung, the remodeling score was calculated as the mean of all scores collected per lung.

Processing of the lungs. Briefly, the lungs were placed into formaldehyde for fixation, paraffin embedded and sections stained with H&E (hematoxylin and eosin) or with Periodic Acid Schiff (PAS). The slides were scanned using the Leica Biosystems SlidePath Gateway and the Leica SCN400 (Leica Microsystems Inc., Buffalo Grove, IL, USA) whole-slide scanning system at the Histopathology Core of Experimental Pathology Research Laboratory (RRID:SCR_017928) at NYU Grossman School of Medicine. The digital dynamic slide images were in OMERO software (OMERO.web 5.19.0. 2007–2022 University of Dundee, Open Microscopy Environment and Glencoe Software, Inc.) and analyzed at 200× magnification.

Examples for the pulmonary vascular scoring are shown in Figure 3: 1—normal, 2—thickened but regular, 3—severe remodeling with irregular layers of smooth muscle cells, 4—very severe remodeling and more irregular arrangement of smooth muscle cells.

Severe remodeling was also determined as the percentage of all scored blood vessels per lung that had severely remodeled walls with disorganized layers of smooth muscle cells (smooth muscle cells in the blood vessel wall that assumed a pattern that differed from the lumen, scores of 3 or 4, Figure 3) [1]. For each lung, severe arterial thickening was calculated by the following formula: 100 ÷ number of all scored blood vessels × number of severely remodeled vessels.

mRNA expression. Total RNA from lung tissue was isolated (RNeasy Mini Kit; QIAGEN Inc, Valencia, CA, USA) and reverse-transcribed using the High-Capacity cDNA Reverse Transcription kit (Applied Biosystems, Foster City, CA, USA). Real-time PCR was performed in duplicate with 20 ng of cDNA using the 7900HT Fast Real-Time PCR system (Applied Biosystems). The qPCR for the detection of the gene expression was performed with SYBR Green (Invitrogen, Grand Island, NY, USA) and primers purchased from Origene (Rockville, MD, USA). *RELMα* (*Retnla*), *RELMβ* (*Retnlb*) and *RELMγ* (*Retnlg*) were assayed using TaqMan Gene expression Assay (Applied Biosystems) based on a FAM labeled probe and the corresponding TaqMan gene expression assay for *β-actin*. The sequences for the primers or probes, respectively, are listed in Table 2. The following conditions were used: 95 °C for 10 min, followed by 40 cycles of 95 °C for 15 s and 60 °C for 1 min, followed by a hold at 4 °C. Raw data were analyzed with SDS Relative Quantification Software version 2.3 (Applied Biosystems) to determine cycle threshold (ΔCt). For each sample, ΔCt values were standardized to the housekeeping gene *β-actin* by calculating 1.98ΔCt ×10,000. Data were expressed as β-actin standardized ΔCt values, or as percent-fold-difference from the mean of the wild-type OVA&PM_2.5_ group (set at 100 U).

Statistical analysis: Statistical analysis and graphs were generated with Prism version 9.2 (Graphpad). Data from multiple groups were analyzed for significant differences using the Kruskal–Wallis test (tie-corrected). Pair-wise comparisons were conducted with the unpaired, two-tailed Mann–Whitney U test. *p* < 0.05 was considered to be statistically significant.

## 5. Conclusions

In conclusion, our study demonstrates that the role of RELMα for the pulmonary hypertension phenotype and molecular markers of right ventricular stress is dependent on the mouse background strain, which in turn determines clearly distinct adaptive immune response types in the lungs.

## Figures and Tables

**Figure 1 ijms-24-11918-f001:**
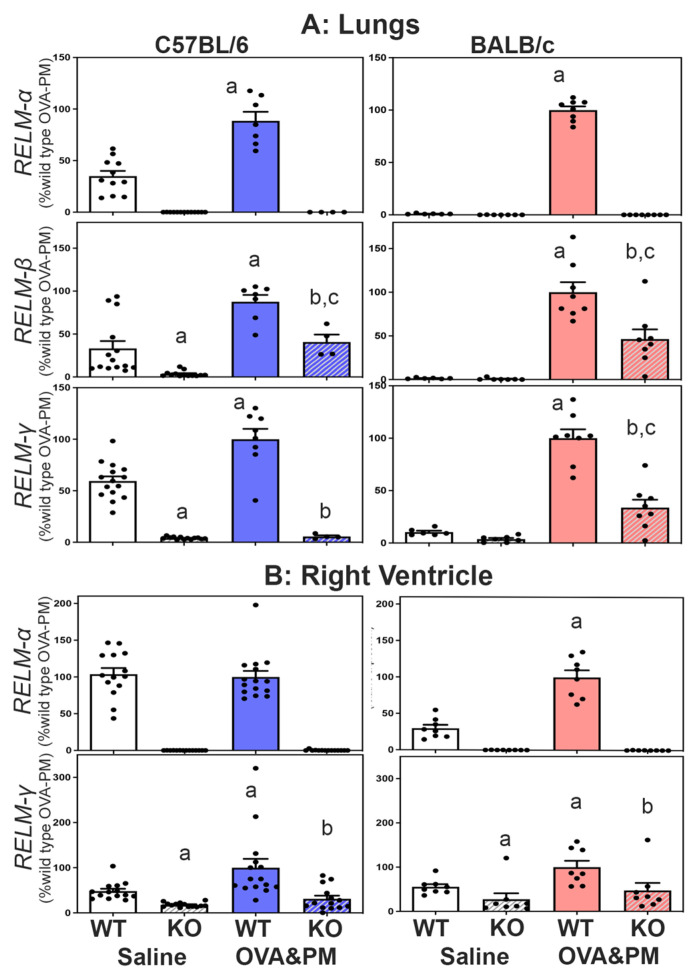
Expression of *RELMα, β* and *γ* mRNA in the lungs (**A**) or the right ventricle (**B**). *RELM β* was not detected in the right ventricle. Bars show means, SEM and individual data points of relative CT values measured to β-actin and calculated relative to the mean of the OVA-PM wild-type groups at 100 U. Groups were compared by independent, 2-tailed Mann–Whitney test, *p* < 0.05 for comparison with (a) WT-Saline, (b) WT-OVA&PM, (c) KO-Saline.

**Figure 2 ijms-24-11918-f002:**
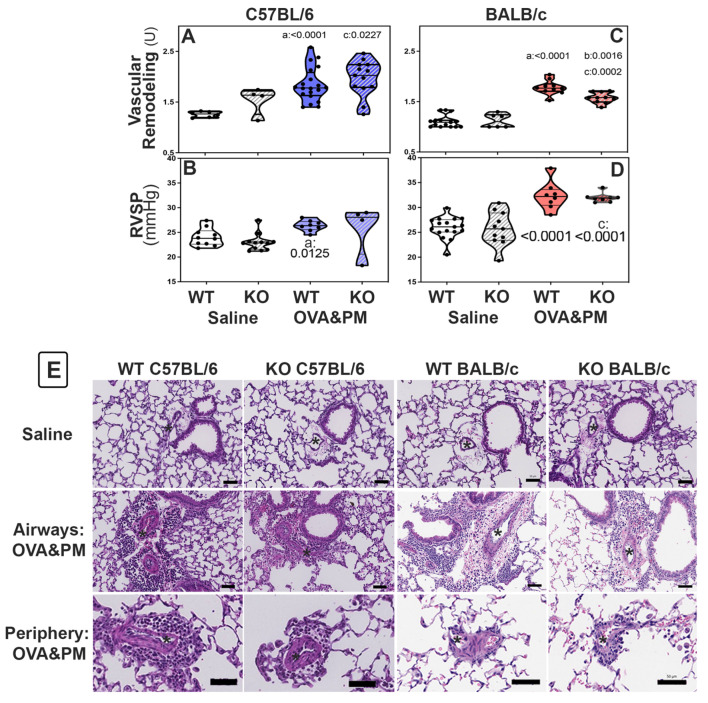
Pulmonary vascular responses to exposure with OVA&PM in C57BL/6 strain (**A**,**B**,**E**) or BALB/c (**C**,**D**,**E**) strain wild-type (WT) or RELMα KO (KO) mice measured by pulmonary arterial remodeling scores (**A,C**,**E**) and right ventricular systolic pressure (RVSP, (**B**,**D**)). Violin plots show individual data points and data distribution. Comparisons were with independent, 2-tailed Mann–Whitney test. *p* < 0.05 for comparison with (a) WT-PBS, (b) WT-OVA&PM and (c) KO-PBS. (**E**) Photomicrographs of Hematoxylin–Eosin stained sections of lungs showing areas surrounding airways or areas in the periphery of the lungs. Scale bars represent 50 μm, * indicate arteries, one example per image.

**Figure 3 ijms-24-11918-f003:**
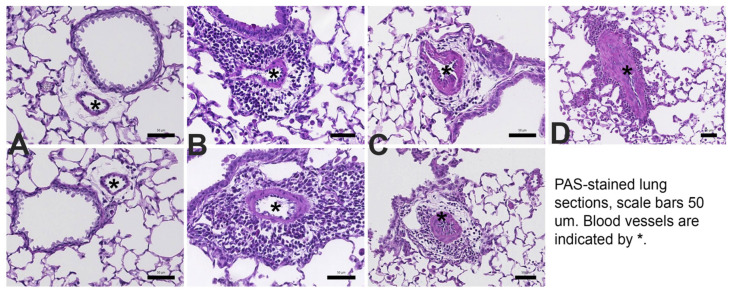
Photomicrographs show examples of small <100 µm pulmonary arteries (*) and the method to determine remodeling scores: (**A**) normal—1; (**B**) mild—2; (**C**) severe—3; (**D**) very severe—4.

**Figure 4 ijms-24-11918-f004:**
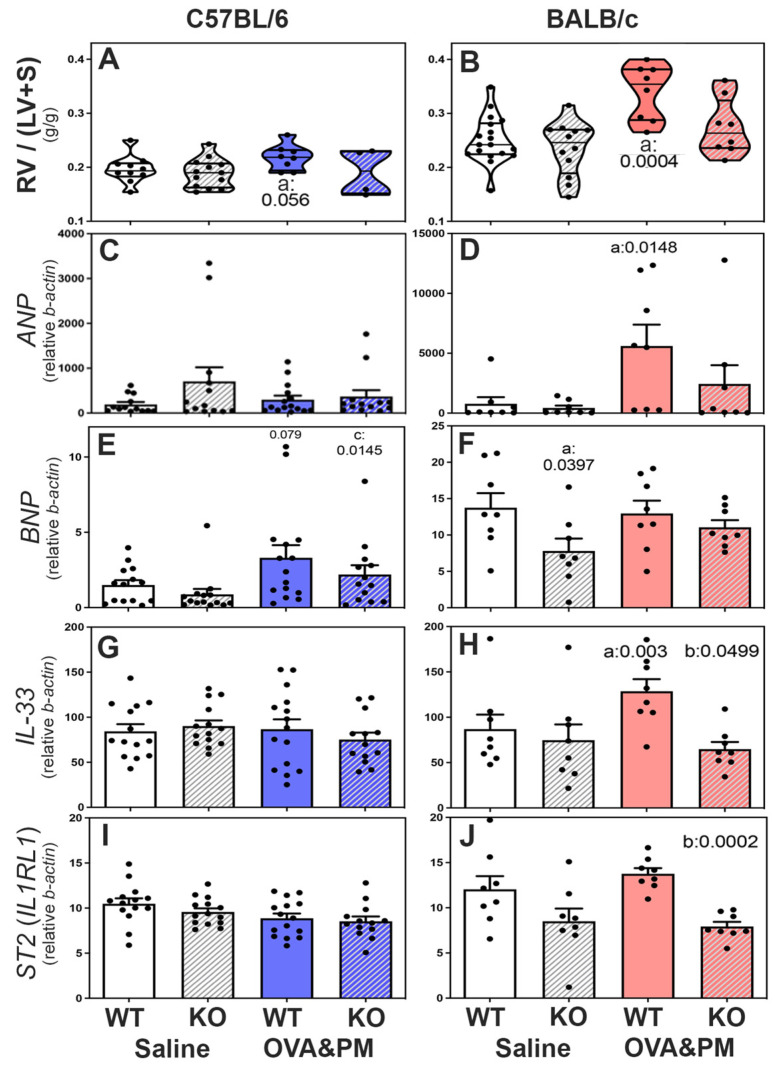
Right ventricular responses in C57BL/6 strain (**A**,**C**,**E**,**G**,**I**) and BALB/c (**B**,**D**,**F**,**H**,**J**) strain WT and RELMα KO mice measured by right ventricular weight (RV/LV + S) × 100 (**A**,**B**) and expression of mRNA of mediators that reflect RV stress (*natriuretic peptide type A-ANP*, or *type B-BNP*, *IL-33*, *ST2-IL1RL1-**IL-33* receptor). Violin plots (**A**,**B**) show individual data points and data distribution. Bars (**C**–**J**) show means, SEM and individual data points. (**C**–**J**): CT values were calculated relative to beta-actin (×1000). Pairwise comparisons were with the independent, 2-tailed Mann–Whitney test; *p* < 0.05 for comparison with (a) WT-Saline, (b) WT-OVA&PM and (c) KO-Saline.

**Figure 5 ijms-24-11918-f005:**
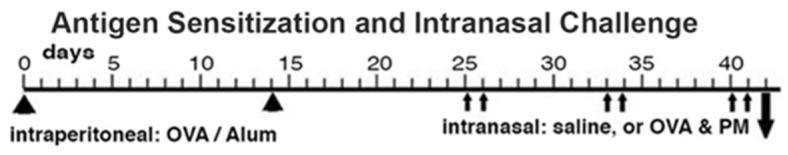
Schematic representation of the sensitization and challenge protocol. The timeline shows the procedures: intraperitoneal injection (arrowheads), intranasal administration (small arrows), terminal analysis and tissue harvest (big arrow).

**Table 1 ijms-24-11918-t001:** Percentage of severely remodeled pulmonary arteries following exposure to OVA-PM ^1^.

Severely Remodeled Pulmonary Artery (OVA-PM Exposed)	C57BL/6				BALB/c			
	Median	Quartiles	Group *n*	*p*-Value	Median	Quartiles	Group *n*	*p*-Value
Wild Type	12.03	1.315, 50.460	8	0.1919	7.275	0.00, 10.390	10	0.0126
RELMα−/−	39.13	19.790, 41.800	9		0.000	0.00, 3.846	10	

^1^ Data represent the percent of severely remodeled arteries (score of 3 and more) in each lung calculated based on the analysis of at least 20 view fields visualized at 200× magnification.

**Table 2 ijms-24-11918-t002:** List of primers and probes.

Target	Gene Name	Sequence (5’ to 3’)
ANP-F ^1^	*nppa*	TACAGTGCGGTGTCCAACACAG
ANP-R	*nppa*	TGCTTCCTCAGTCTGCTCACTC
BNP-F	*nppb*	TCCTAGCCAGTCTCCAGAGCAA
BNP-R	*nppb*	GGTCCTTCAAGAGCTGTCTCTG
*IL-33*-F	*il33*	ACTGCATGAGACTCCGTTCTG
*IL-33*-R	*il33*	CCTAGAATCCCGTGGATAGGC
ST2 F	*il1rl1*	GGATTGAGGTTGCTCTGTTCTGG
ST2 R	*il1rl1*	TCGGGCAGAGTGTGGTGAACAA
β-actin-F	*actb*	GGCTGTATTCCCCTCCATCG
β-actin-R	*actb*	CCAGTTGGTAACAATGCCATGT
RELMα (TaqMan)	*retnla*	CTTGCCAATCCAGCTAACTATCCCT
RELMβ (TaqMan)	*retnlb*	GGAAGCTCTCAGTCGTCAAGAGCCT
RELMγ (TaqMan)	*retnlg*	AAACCTGGCTCATATCCCATTGATG
Actin, β (TaqMan)	*actb*	ACTGAGCTGCGTTTTACACCCTTTC

^1^ F—forward, R—reverse.

## Data Availability

Data supporting the reported results can be found in the manuscript. Additional details may be requested from the corresponding author Grunig.

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
