# Peer review of "Resistin-like Molecule α and Pulmonary Vascular Remodeling: A Multi-Strain Murine Model of Antigen and Urban Ambient Particulate Matter Co-Exposure"

_ijms, 2023, doi:10.3390/ijms241511918_

Round 1

Reviewer 1 Report

Please more clearly state the conclusion of this study in terms of the role of RELMalpha in pulmonary vascular remodeling in the Abstract.

Please show representative histology images of PA and RV for Saline WT, Saline KO, OVA-PM WT, and OVA-PM KO.

Please provide quantifications of the PA wall thickness.

Author Response

The authors thank the reviewer for the careful and constructive comments that improved our manuscript. Please find the point-by-point answers below in gray background. In the manuscript, changes are marked on the margin of the text.

Please more clearly state the conclusion of this study in terms of the role of RELMalpha in pulmonary vascular remodeling in the Abstract. - DONE

Please show representative histology images of PA and RV for Saline WT, Saline KO, OVA-PM WT, and OVA-PM KO. - DONE

Please provide quantifications of the PA wall thickness. - 

The PA remodeling was quantified by scoring of all arteries in at least 20 view fields per lung, and the mean score calculated per mouse.  This is better explained in the methods section.

Simply quantifying the wall thickness would describe the pathological changes in the PA, if the remodeling was concentric (score of 2 in our mouse lungs). However, antigen-PM2.5 exposure induces irregular reorganization of the smooth muscle cells (scores of 3 or 4 in the mouse lungs), as we have reported (Daley et al. J Exp Med 2008. 205, 361-72 and Grunig et al. Pulm Circ, 2014. 4: 25-35).  We have explained this better in the methods section with examples in figure 2, and adding figure 4e that shows the type of remodeling changes much better.

Reviewer 2 Report

In their experimental work, the authors investigated the pathomechanism of changes leading to remodeling in pulmonary hypertension. Such studies are particularly important due to the difficulties faced by the practitioner treating patients with PAH. New metabolic dishes may translate into new therapeutic options...
Work well planned, results clear, well discussed.
I have a few suggestions for the introduction, although they can be implemented in the discussion as well:
- please add a few sentences about other trails (authors basically limit themselves to cutokins). The role of microRNA and lncRNA
- soluble guanylyl cyclase activators as therapeutic option in the pharmacotherapy of heart failure and pulmonary hypertension
- what is the role of carbon monoxide and nitric oxide as examples of the youngest class of transmitters
- what are the vasodilating properties of human mesenteric arteries constricted with endothelin-1 in animal model
- what are the interactions of these pathways with those observed in the study. This is important because of the pathways whose inhibition is currently used in clinical medicine

Author Response

The authors thank the reviewer for the careful and constructive comments that have significantly improved our manuscript. Please find the point-by-point response below, our answers are in grey background.

  • please add a few sentences about other trails (authors basically limit themselves to cutokins). The role of microRNA and lncRNA - DONE, discussion, please see below. In the manuscript, changed paragraphs are marked on the side of the text.

  • - soluble guanylyl cyclase activators as therapeutic option in the pharmacotherapy of heart failure and pulmonary hypertension - DONE, introduction, please see below

  • - what is the role of carbon monoxide and nitric oxide as examples of the youngest class of transmitters - DONE, introduction, please see below

  • - what are the vasodilating properties of human mesenteric arteries constricted with endothelin-1 in animal model - DONE, introduction, please see below

  • - what are the interactions of these pathways with those observed in the study. This is important because of the pathways whose inhibition is currently used in clinical medicine DONE, discussion, please see below

In the introduction we have discussed currently used therapeutic agents and their ability to induce vasodilation and some anti-proliferative effects. The types of mediators like RELMa investigated in our current study are potential modulators of the pathogenic processes that cause the pulmonary vascular changes, and therefore complement and add to the vasodilative agents.

In the discussion section, we have carefully discussed other types of mediators (e.s. epigenetic markers, miRNA, lncRNA, or agonists of BMPR2 receptor activity). And we discussed how current therapies could interact with potential future anti-cytokine drugs.

Reviewer 3 Report

This is an animal study, regarding RELMα and pulmonary vascular remodeling in hypoxia-induced PH. The authors concluded that the role of RELMα for pulmonary hypertension phenotype and molecular markers of right ventricular stress was dependent on the mouse background strain that in turn determines clearly distinct adaptive immune response types in the lungs. Although the authors reported the difference between the mice background, the mechanisms should be clarified in near future. 

This reviewer considers that the authors well performed the present study, and has some comments as described below. 

Major comments:

1.     The aim of the present study was not clear. The authors should clearly describe the aim of the study at the end of the Introduction section.  

2.     Figure 3. Were the expression of RELMα, β, and γ mRNA? The authors should clearly mention this issue. 

3.     Although the authors showed the photomicrograph examples of small pulmonary arteries in Figure 2, but they should also show the demonstrable pictures of small pulmonary arteries from the results of the present study, combined with Figure 4.  

4.     Figure 5. In general, ventricular stress is reflected as BNP, rather than ANP. Why did the authors examine ANP? The authors should mention this issue. 

Author Response

The authors thank the reviewer for the careful and constructive comments that have strengthened the manuscript. Please find the point-to-point response below, the answers are in grey background. In the manuscript changed paragraphs are marked on the side of the text.

Major comments:

  1. The aim of the present study was not clear. The authors should clearly describe the aim of the study at the end of the Introduction section.  

Done

2. Figure 3. Were the expression of RELMα, β, and γ mRNA? The authors should clearly mention this issue. Yes, done

3. Although the authors showed the photomicrograph examples of small pulmonary arteries in Figure 2, but they should also show the demonstrable pictures of small pulmonary arteries from the results of the present study, combined with Figure 4.  Done

4. Figure 5. In general, ventricular stress is reflected as BNP, rather than ANP. Why did the authors examine ANP? The authors should mention this issue. We added the data for BNP expression to Figure 5. BNP mRNA expression was increased in the RV of C57BL/6 mice exposed to OVA-PM, but not in BALB/c mice.

Round 2

Reviewer 1 Report

.

Reviewer 3 Report

This reviewer has no further comment.